# Factors Affecting Sugarcane Production by Small-Scale Growers in Ndwedwe Local Unicipality, South Africa

**Nkosingiphile Samuel Zulu, Melusi Sibanda * and Bokang Stephen Tlali**

Department of Agriculture, University of Zululand, KwaDlangezwa 3886, South Africa
* Correspondence: SibandaM@unizulu.ac.za; Tel.: +27-(0)-35-902-6068

**Abstract:** Sugarcane is an important crop worldwide due to its many nutritional and economic uses. Small-scale sugarcane growers (SSGs) are a significant sector of sugarcane production in South Africa. However, the number of SSGs is noted to have declined from as early as the 2000s to the present time. As a result of the declining sugarcane production, there are now generally fewer SSGs. However, it is not clear cut as to what caused the decrease in sugarcane production by SSGs. The primary objective of this paper is to determine the factors affecting the sugarcane production by SSGs in Mona and Sonkombo in Ndwedwe Local Municipality. Data collection was through a well-structured questionnaire administered to 100 SSGs (that is, 50 respondents each from the study sites, namely Mona and Sonkombo) that were randomly selected. The paper employs descriptive statistics to describe farm characteristics, and a production function (Cobb–Douglass production function (CDPF)) analysis using the ordinary least squares (OLS) criterion to estimate the parameters affecting sugarcane production. Results show that late harvesting (by up to three (3) weeks), late fertiliser application (by up to six (6) months, and chemicals (Gramoxone) application (by up to five (5) months) were primary challenges facing SSGs, likely to result in declining sugarcane yield. The CDPF regression analysis reveals that significant predictors of the production function are: labour and the amount of chemicals (Gramoxone) applied. Labour (man-days/ha), amount of chemicals (Gramoxone) applied are found to be statistically significant and positively correlated with sugarcane production. The government, through the relevant Department of Agriculture, including the private sector, should intensify out-grower technical services for SSGs to realise higher production per hectare. Such services would ensure optimal allocation and application of inputs, labour and chemicals (herbicides and pesticides), respectively, at the right time to ensure efficacy. There is also a need to introduce buying consortiums for SSGs to reduce the costs of inputs.

**Keywords:** Cobb–Douglass production function; cane growers; yield; KwaZulu-Natal

## 1. Introduction

Sugarcane is regarded as an essential crop worldwide due to its extensive use in the day-to-day lives of people and its industrial use intended for dietary and economic sustenance [1]. Sugarcane is also an important industrial crop of subtropical and tropical regions worldwide. According to the Fair Labor Association [2], approximately 28.3 million hectares are planted with sugarcane in more than 90 countries with a total production of about 1.69 billion tonnes worldwide.

The sugar industry in South Africa has been reported as an industry with a high socio-economic developmental focus in rural areas by organising resources, creating job opportunities, providing a source of income and developing transport and communication networks [3]. However, Garside and Bell [4] state that although there are benefits obtainable from sugarcane production, the sugar

industry has experienced various challenges, which are encountered mainly by SSGs. Dubb [5] reveals that in South Africa, the sugar industry has been facing a problem of declining sugarcane production, particularly by SSGs. The decline of sugarcane production yield by SSGs causes distress to the South African sugar industry [4]. These challenges facing SSGs have affected their productivity, and as a result, the industry's earnings have dropped over the years.

As a result, this has led to fewer SSGs and also limiting their ability to commercialise. The decline in sugarcane production by SSGs has also increased reliance on government social grants, for example, old-age pensions and child support, due to low net-farm income. Although it is common knowledge that the number of farmers has decreased because of the resultant lower production and farm income, it is unclear what has caused the decrease in production which has led to low income and eventually fewer farmers.

A preliminary report on a survey conducted in Mauritius and South Africa in 2009 indicates that poor re-planting rates and weeds contributes to reduced yields, and low levels of education add to poor crop husbandry practices among SSGs [6]. Eweg et al. [6] also report that SSGs perceive weeds to be the top agronomic constraint. According to Conlong and Campbell [7], improving weed management practices amongst SSGs in the South African sugar industry needs attention, because weeds are assumed to be another cause of yield decline.

Crop protection practices such as the use of herbicides amongst SSGs also need to be addressed. Eweg et al. [6] state that high costs of inputs such as fertiliser and chemicals may be one of the significant constraints affecting SSGs' yields, putting a strain on profit margins as sugar prices have not kept pace, implying that SSGs do not apply enough inputs such as straight and mixed fertiliser.

Another difficulty experienced by SSGs is the adjustment of the minimum wage as regulated by labour law, which has raised the financial and improbability costs to farmers to pay workers [8]. A high percentage increase in wages paid to farmworkers will have a significant impact on SSGs' net farm income. Moreover, there is some indication suggesting that before the minimum wage adjustment, farmers may have been over-employing unproductive labour; after which the law came into effect, they have been required to employ fewer skilled workers, working many hours a day.

According to Conlong and Campbell [7] the rising input costs for sugarcane growing in KwaZulu-Natal, particularly in the planting areas of Ntumeni and Showe, are resulting in less profit for SSGs. The consequences of rising input costs influence the performance and progression of the industry. Small-scale sugarcane growers, therefore, need to find ways to reduce the effects of increasing input costs.

Mandla, Mnisi and Dlamini [9] suggest that SSGs have a complicated relationship with financial institutions, upon whom they depend for working capital to support their sugarcane fields. Many other factors impact the production of SSGs in KwaZulu-Natal [9], including drought, farm size, inadequate infrastructure, low educational attainment and unskilled labour [3].

Dubb [5] reveals that from the late 1970s to the early 1980s, SSGs' production was camouflaged as microcredit that introduced Bantustan land into sugarcane production under a sturdy administration of mills. Dubb [5] further states that in the late 1980s towards early 1990s, these subsidies were removed, which encouraged millers to subcontract farmer support. The millers concurrently introduced a rise in SSG figures by eliminating limitations on grower registration [10].

A particular disadvantage of the SSGs is that they lack adaptive strategies. Mandla et al. [9] note that SSGs' production had been declining at an alarming rate, for example, from about 57,000 SSGs in 2000 to fewer than 14,000 SSGs in 2011. Dubb [5] attributes this decline in sugarcane production in KwaZulu-Natal to extreme climate events such as drought since SSGs lack adaptation strategies. In contrast, commercial (large-scale) growers such as Tongaat-Hulett's mill have not suffered from the effect of drought because they have adaptive strategies such as irrigation schemes. However, the declining sugarcane production yield by SSGs cannot be attributed to climate change alone but also to poor agronomic practices, and numerous factors [11] that still need further investigation.

Although numerous studies have researched the reasons for the decline in sugarcane production, particularly for large-scale farmers, the need for further research remains, more so for SSGs. This necessity is because sugarcane production is still on the decline, resulting in fewer SSGs. The aim of this paper, therefore, is to determine the factors affecting sugarcane production by SSGs in Mona and Sonkombo within Ndwedwe Local Municipality of KwaZulu-Natal Province. The precise objectives of the paper are to (1) describe the challenges facing SSGs, and (2) determine the input-output relationship in sugarcane production by SSGs through a mathematical production function. Such an understanding of the input-output relationship in sugarcane production can help SSGs to use inputs in production to achieve optimal sugarcane yield efficiently. Governments can also effectively use the information to come up with policies that will assist SSGs in improving their sugarcane production, given the challenges facing SSGs.

## 2. Materials and Methods

### 2.1. Selection of Study Sites

This paper presents the findings of an investigation in Ndwedwe Local Municipality of iLembe District Municipality of KwaZulu-Natal Province. Ndwedwe is one of the four local authorities that form part of iLembe District Municipality [12]. In broad terms, Ndwedwe Local Municipality lies approximately 20 km inland from the Kwa-Zulu Natal coast and borders the east of KwaDukuza Local Municipality and the north of Maphumulo Local Municipality [12]. The specific study sites are Mona and Sonkombo villages. We purposively select Mona and Sonkombo from a total of five villages planted with sugarcane in the local municipality, the other three being Ndwedwe Mission, Ntaphuka, and Nhlangano. Mona and Sonkombo are chosen based on the level of sugarcane production (with sugarcane being the leading crop enterprise), and these areas have identical agro-climatic conditions. Again, both Mona and Sonkombo have SSGs who produce and deliver sugarcane to the mill. Figure 1 is a map showing the location of study sites: Mona and Sonkombo in Ndwedwe Local Municipality.

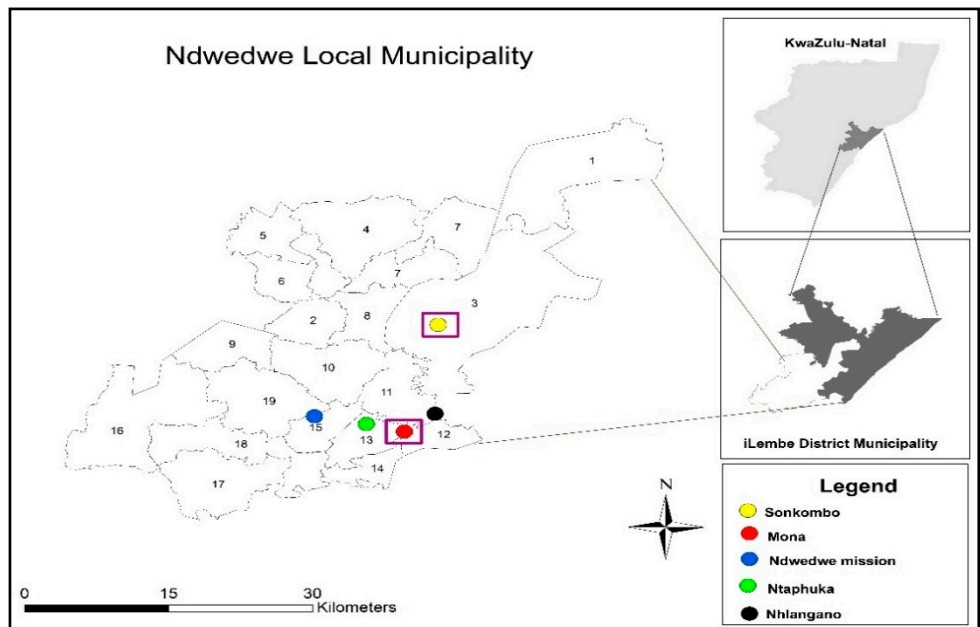

**Figure 1.** Map showing the location of Mona and Sonkombo in Ndwedwe Local Municipality.

### 2.2. Research Design

Following a quantitative research methodology, a descriptive cross-sectional research design is used in this paper to describe the challenges and determine the factors affecting sugarcane production by SSGs in Ndwedwe Local Municipality in KwaZulu-Natal. Chidoko and Chimwai [13] state that a

descriptive survey design details and clarifies the existing achievements, attitudes, behaviours and other characteristics of the group of subjects. A quantitative method, therefore, can provide a high level of measurement precision and statistical power and eliminates the bias of judgment.

Conceptual Framework on Factors Affecting Sugarcane Production by Small-Scale Growers

To identify and define variables (factors) affecting sugarcane production by SSGs for this paper, the authors first need to decompose the factors affecting agricultural production in general. It is essential to mention that productivity is not an outright measure, but rather an estimation of the relationship of the input/output ratio. From an economic standpoint, a standard model that explains the input/output relationship in agriculture is the production function model. The underlying assumption is that the availability of resources determines production by any farmer. More so, the primary factors of agricultural output comprise land, labour and capital [14]. From literature, numerous factors have been shown to affect agricultural productivity by either an increase or decrease in it. These factors include socioeconomic (human assets), institutional (financial assets), physical (tangible assets), and environmental (natural assets) factors. Studies show that these various factors have any effect on agricultural production, which we proxy by yield—tonnes/ha in this paper. The conceptual framework on factors affecting sugarcane production by SSGs in this paper, therefore, is grounded on the production function model, encompassing the factors revealed in literature. This paper acknowledges that from the literature review, other socio-economic factors (human assets like age, education level, gender, marital status, and extension support; financial assets like non-farm income, access to credit, farm income) and exogenous factors (natural assets), for example, water and climatic factors, influence agricultural production. However, due to data constraints and the complexity of measuring and directly linking these variables to the production process, this paper focuses only on the crop management practices that include natural assets (cultivated land size), human assets (labour), and physical assets (fertiliser, urea, and chemicals applied). Figure 2 is the conceptual framework (illustration) of how the identified factors in this paper affect sugarcane production in Mona and Sonkombo.

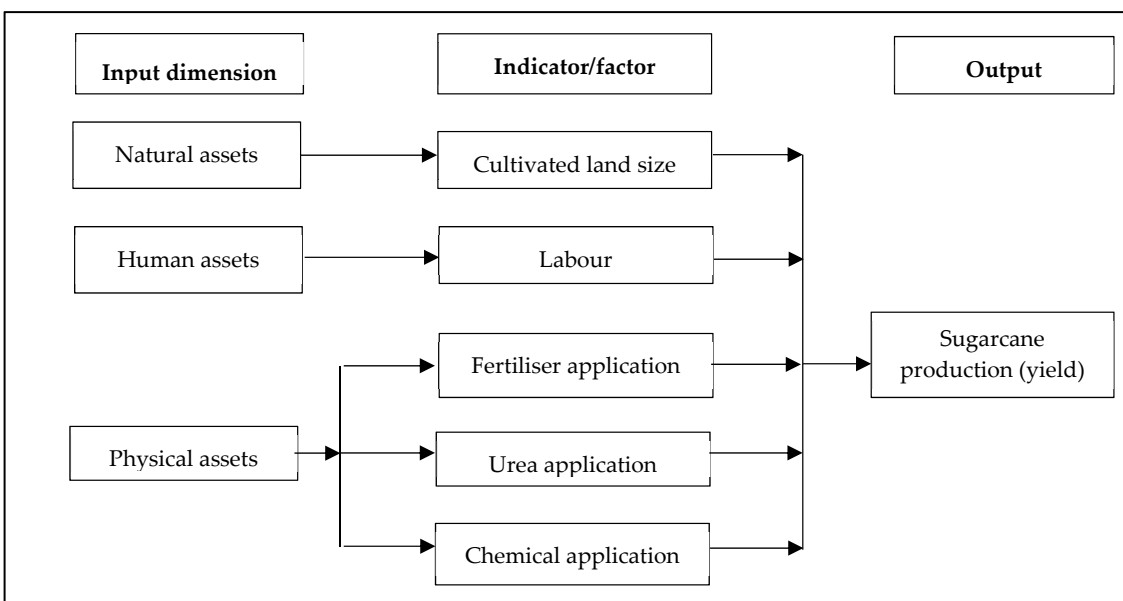

**Figure 2.** Conceptual framework of the factors affecting sugarcane production in Mona and Sonkombo.

*2.3. Sampling Size and Procedures*

According to Bless et al. [15], the use of a complete and correct sampling frame is the first means of ensuring a representative sample. A list of all the SSGs who delivered sugarcane to the mill at the time of the study is available from the Tongaat-Hulett office of Nhlangano region. There are approximately

2000 SSGs (that is about 400 SSGs from each of the five villages) that plant sugarcane for marketing or crushing in Ndwedwe Local Municipality [16]. The sample size of the results presented in this paper is an estimation from this sampling frame. For a sample to be scientific, it is not only about how many elements should be included for observation. It requires careful thinking about which items to add and how to select them for probability sampling [17]. This research was carried out through multistage sampling whereby SSGs who deliver sugarcane to the mill are chosen first purposively with the assistance from extension workers in Mona and Sonkombo. Secondly, SSGs that deliver sugarcane to the mill is then picked randomly. Tustin et al. [18] state that the chance of selecting respondents with simple random technique in a sample is known and the same for the entire population studied. In total, the sample size in this paper consists of 100 SSGs (that is 50 SSGs from each of the two villages—Mona and Sonkombo, respectively). In as much as this sample size is arguably small to generate robust results and statistical power of the mathematical model applied in this paper, it is a reasonably accessible sample in terms of time and costs.

### 2.4. Data Collection

Data collection for this research adopted a well-structured questionnaire composed of open-ended and close-ended questions. The survey borrows on the work of Hussain and Khattak [19], who did a similar study on the economics of sugarcane production in the Charsadda District of Pakistan. Hussain and Khattak [19] established that the significant predictors affecting sugarcane production include cultivated area, amount of fertiliser applied, amount of chemicals (pesticides and insecticides) applied, labour (human and mechanical—tractor) and amount of seed used. The questionnaire was piloted on SSGs that did not constitute the final sample. The questionnaire was revised using responses acquired in the pre-test. We translated the questionnaire from English to isiZulu—the native language in the study area. The University of Zululand Research Ethics Committee granted ethical clearance for this research before it commenced. The research adhered to the policies of the university on research procedures and research ethics. The questionnaire administration to respondents was through face-to-face interviews. Bless [15] states that a face-to-face administered questionnaire is possible with respondents who can neither read nor write. Besides, the existence of the interviewer raises the excellence of the responses, as the interviewer can search for more specific answers.

### 2.5. Data Analysis

Data analysis for this paper uses descriptive and inferential statistics. Data coding, capturing, and transformations were through Microsoft Excel version 2016 (Microsoft Corporation, Washington, DC, USA). Entering and editing data on Microsoft Excel is relatively straightforward. The dataset created in Microsoft Excel is exportable to the Statistical Package for Social Sciences (SPSS) version 25 (SPSS Inc. (IBM), Chicago, IL, USA) for analysis. Descriptive statistics performed comprise the frequencies, percentages, means, range (minimum and maximum) and standard deviations. Here, the information is tested for validity. Specifically, in describing the farm characteristics, and production challenges, descriptive statistics is applied. Additionally, a production function; the Cobb–Douglas production function (CDPF) is used for inferential analysis—to determine the factors affecting the sugarcane production by SSGs employing an ordinary least squares (OLS) criterion. The information is interpreted and presented in the form of tables.

### 2.5.1. Specification of the Inferential Analysis

A CDPF is used to estimate the factors affecting sugarcane production by SSGs in this paper. The Cobb–Douglas form of the production function is broadly used in economics to describe the input/output relationship [20]. Numerous similar studies have applied the CDPF, for example, Wongnaa [14]; Baiyegunhi and Arnold [20]; Ekbom [21]; and Nazir et al. [22]. That motivates the selection of CDPF in this paper, and it does not introduce many econometric estimation challenges,

for example, multicollinearity and heteroscedasticity. The CDPF is also easy to compute, with some properties of flexibility. The CDPF in its stochastic form is specified as follows (Equation (1)):

$$Y = \beta_1 X_{2i}^{\beta 2} X_{3i}^{\beta 3} e^{\mu i} \tag{1}$$

where $Y$ = Output; $X_2$ = Labour input; $X_3$ = Capital input; $\mu$ = Stochastic disturbance term; $e$ = Base of natural logarithm

From Equation (1), the relationship between output and the two inputs is non-linear. The CDPF is a double-log production function, and in this paper, we proxy sugarcane production in yield—tonnes per hectare as the dependent variable. Given the accumulative input prices in the sugarcane industry, the objective is to reduce costs, but maximise production. Therefore, we had to log-transform the estimation. The specification of the log-transformed model is as follows (Equation (2)):

$$\begin{aligned} \text{In}Y &= \text{In}\beta_1 + \beta_2 \text{In}X_2 + \beta_3 \text{In}X_3 + \mu_i \\ &= \beta_0 + \beta_2 \text{In}X_2 + \beta_3 \text{In}X_3 + \mu_i \end{aligned} \tag{2}$$

Thus, the resultant model is linear in parameters ($\beta_0$, $\beta_2$ and $\beta_3$). The CDPF has to be linear for ease of interpretation. The coefficients of the variables inputted in the CDPF are estimated using the OLS technique. The resultant functional econometric model is as follows (Equation (3)):

$$\text{In}Y = \beta_0 + \beta_1 \text{In}X_1 + \beta_2 \text{In}X_2 \ldots + \beta_{13} \text{In}X_{13} \ldots \mu \tag{3}$$

where $\text{In}Y$ depicts the Log of the production per unit, and $\text{In}X_1$ to $\text{In}X_n$ the natural log of the independent continuous variables. The explanatory variables included in the CDPF relies on similar studies; for example, by Hussain and Khattak [19], who investigated "the economics of sugarcane production in Charsadda District of Pakistan"; Baiyegunhi and Arnold [20] who identified the "Economics of sugarcane production on large-scale farms in the Eshowe/Entumeni areas of KwaZulu-Natal, South Africa" and Nazir et al. [22] who identified "Factors Affecting Sugarcane Production in Pakistan"; where; $Yi$ = Sugarcane production (yield in tonnes/ha); $\beta_0$ = Constant; $\beta_i$ = Output elasticities; $X_1$ = Cultivated farm size (ha); $X_2$ = Labour (man-days/ha); $X_3$ = Amount of basal fertiliser applied (kg/ha); $X_4$ = Amount of urea applied (kg/ha); $X_5$ = Amount of chemicals (Gramoxone) applied (litres/ha); $e$ = error term.

2.5.2. Explanation of Variables Used in the Cobb–Douglass Production Function (CDPF) and Their Expected Outcomes

Table 1 is a summary of the explanatory variables used in the CDPF regression analysis and their expected outcomes. Cultivated farm size in this paper refers to the land that is devoted to sugarcane growing measured in hectares. In the case of SSGs, the assumption is that they have a limited managerial capacity of producing in their farms. Small-scale sugarcane growers, therefore, can only manage small-scale farming provided that they have the proper skills to ensure that the relationship between cultivated farmland and production is maintained through applying the correct amount of inputs which are per the size of their farms. The South African Cane Growers Association (SACGA) [23], reports that farm size may not influence production, but rather production may be influenced by input costs, transport costs, and replant rates. Emana and Gebremedhin [24], states that the relationship between the size of the farm and crop production (yield—tonnes/ha) cannot be pre-determined. Therefore, either a positive or negative correlation between the cultivated farm size and sugarcane production (yield—tonnes/ha) by SSGs is expected.

**Table 1.** Explanatory variables used in the regression analysis and their expected outcomes.

| Parameter | Description and Measurement | Data Type | Expected Outcome |
|---|---|---|---|
| Cultivated farm size | The proportion of farmland devoted to farming land (ha) | Continuous | + |
| Labour | Amount of labour required for various operations (man-days/ha) | Continuous | + |
| Amount of basal fertiliser applied | Amount of basal fertiliser applied during planting of sugarcane (kg/ha) | Continuous | + |
| Amount of Urea applied | Amount of urea applied for top-dressing (kg/ha) | Continuous | + |
| Amount of chemicals (Gramoxone) applied | Amount of chemicals (Gramoxone) applied in the field (litres/ha) | Continuous | + |

+; ha; kg; denotes positive or negative association with the dependent variable; hectares; and kilogram respectively.

The South African Cane Growers Association [23], states that the availability of labour on the farm is increasingly becoming a limiting factor in the sugarcane industry for both commercial and small-scale farmers. Sugarcane production is labour demanding for many operations within the growing season. Therefore, to attain a good crop; a significant amount of work is required. However, labourers must have the technical know-how in their operations for them to be able to produce good-quality crops. Given that most of the SSGs use contractual labour, this paper expects a positive correlation between labour (man-days/ha) and sugarcane production (yield—tonnes/ha) by SSGs.

A sufficient amount of basal fertiliser is imperative for the production of a quality crop [25]. It is crucial that optimal amounts of fertiliser are applied for the sugarcane crop to have all the essential nutrients to obtain a good yield. However, fertiliser (measured in kg/ha) application has to be per the soil recommendations. A positive correlation is likely between the amount of fertiliser applied, and sugarcane production (yield—tonnes/ha) provided farmers to adhere to soil recommendations [19].

The top-dressing of land, according to the recommended fertiliser requirements, is essential for the sugarcane crop's vegetative growth [26]. The application of urea (kg/ha) for top-dressing boosts the vegetative sugarcane growth. This paper expects a positive correlation between the amount of urea applied and sugarcane production (yield—tonnes/ha) by SSGs.

Chemicals (such as Gramoxone used as a proxy for chemicals in this paper, which is postemergence and lasso which is the pre-emergence herbicide) application is necessary for crop protection against pests, diseases and as a weed control method. The removal of weeds can lead to the removal of pests and diseases in the field. However, correct calibration of chemicals is paramount. An adequate amount of chemicals (litres/ha) applied in the field is imperative to achieve a good sugarcane yield [19]. This paper expects a positive correlation between the amount of chemicals (Gramoxone) applied and sugarcane production (yield—tonnes/ha) by SSGs.

## 3. Results

### 3.1. Farm Characteristics of the Interviewed Sugarcane Small-Scale Growers in Mona and Sonkombo

This section presents findings on the farm characteristics of the SSGs. Concerning the estimated cultivated land sizes, results reveal that the interviewed SSGs in Mona and Sonkombo on average cultivate a mean land size of 1.21 ha, with a minimum and maximum yield of 0.1 and about 8 ha respectively (Table 2). The results in this paper suggest that SSGs in Mona and Sonkombo own small pieces of land where they produce sugarcane. Sugarcane production is arduous, and high amounts of skilled labour are necessary for various operations during the production season [27]. The results in this paper show that the mean labour for SSGs in Mona and Sonkombo for sugarcane production is 1.66 man-days/ha and ranges from about 1 to 11 man-days/ha. Qualitative data indicate that SSGs relied primarily on contractors to do land preparation, planting, gab filling, fertiliser and herbicide application, hand weeding and harvesting of sugarcane. The mean labour man-days/ha is slightly

lower than the industry standards of 2.5 man-days/ha for example for weed control using knapsack sprayer and four (4) man-days/ha for land preparation to planting [28].

**Table 2.** Summary of farm characteristics of the interviewed sugarcane small-scale growers (*n* = 100).

| Parameter | Descriptive Statistic | | | |
|---|---|---|---|---|
| | **Mean** | **Std. Dev** [1] | **Min** [2] | **Max** [3] |
| Cultivated farm size (ha) | 1.21 | 1.05 | 0.1 | 8.18 |
| Labour (man-days/ha) | 1.66 | 1.93 | 0.5 | 11 |
| Amount of basal fertilizer applied (kg/ha) | 325.50 | 262.42 | 50 | 2050 |
| Amount of Urea applied (kg/ha) | 322 | 259.69 | 50 | 2050 |
| Amount of chemicals (Gramoxone) applied (litres/ha) | 559 | 663.57 | 50 | 1600 |
| Sugarcane yield (tonnes/ha) | 60.83 | 52.45 | 5 | 409 |

[1] Standard deviation, [2] Minimum, [3] Maximum.

Correct basal fertiliser, urea and chemicals (Gramoxone) application can increase sugarcane production. Table 2 shows that the mean amount of basal fertiliser (fertiliser mixture of 5:1:5 (45)) applied by the interviewed SSGs at the time of the study is about 326 kg/ha and ranges from 50 to 2050 kg/ha. The results show that the mean amount of basal fertiliser applied is lower than the industry estimated standards of fertiliser use (600 kg/ha) as per the soil recommendations of the study area.

Concerning urea application, results show that the mean amount of urea applied is about 322 kg/ha and ranges from 50 to 2050 kg/ha (Table 2). The results show that the mean amount of urea applied by the interviewed SSGs is somehow within the industry estimated standards of about 250 kg/ha.

Hogarth and Allsopp [29] state that some weeds release compounds that are toxic to sugarcane growth. Therefore, chemicals application such as herbicides can be useful and economical in sugarcane production [30]. The results show that the mean amount of chemicals (Gramoxone) applied is 559 L/ha and ranges from 50 to 1600 L/ha. The results also show that the mean amount of chemicals (Gramoxone) applied by the interviewed SSGs is somehow within the industry estimated standards of about 400 L/ha.

Concerning the yield of sugarcane, Table 2 shows that mean yield attained by the interviewed SSGs is about 61 t/ha, with a minimum and maximum yield of 5 and 409 t/ha respectively.

*3.2. Production Challenges Faced by Sugarcane Small-Scale Growers in Mona and Sonkombo*

Generally, in South Africa, the small-scale agricultural sector has been discouraged by the lack of access to financial assistance such as operational loans necessary for the sustenance of agricultural production [31]. However, the sugarcane industry is fortunate to have had financial support for more than 50 years and, currently, that holds the deliveries to the mills as a security mechanism [1]. Despite this support, SSGs still face challenges in their production. Table 3 is a summary of the main challenges indicated by the interviewed SSGs. Results show that all (100%) of the interviewed SSGs agree that they experience late harvesting (delays in transportation from the field to loading zone and the sugar mill, immature sugarcane burning and sugarcane being left in the field, resulting in livestock encroachment before and after harvesting) (Table 3). Qualitative data reveal that late harvesting was up to three (3) weeks. According to the sugar industry recommendations, sugarcane must be burnt to cut within one (1) day to avoid deterioration of sucrose. If late harvesting occurs, the average purity loss estimates are about 1.25% per day, that is 2% recoverable value (RV) or 2.2% sucrose loss. Cockburn et al. [32] mention that SSGs perceive weeds as the top agronomic constraint. Table 3 shows that a higher proportion (97%) of the interviewed SSGs agree that weed control is inadequate because it is often late. The recommendation is that chemical application pre- and post- the emergence of herbicides be timely (as soon as the removal of the last stack of sugarcane to control weeds effectively). Qualitative data reveal that the application of the chemicals by SSGs was late by up to five (5) months. The weeds control, which is very late, is expected to compete with the sugarcane crop, thus affecting its productivity. According to Ogwang [33], for efficient production, the selection of the plant material and

its management is one of the crucial elements. The majority (72%) of the interviewed SSGs point out that they use planting material from other growers who are assumed to manage it appropriately and check for diseases (Table 3). The challenge here is the lack of farmer involvement in the selection of the seed variety. Concerning crop nutrition, questions were asked to extract information on the soil tests and fertiliser use decision-making processes. A higher proportion (70%) of the interviewed SSGs indicate that they never did soil analysis in their field, but they plant sugarcane without using any soil recommendations (Table 3). The resultant here is low and late fertilisation (by up to six (6) months) post- or at the planting of sugarcane. Chemicals application; pre and post herbicide application in conjunction with mechanical cultivation, can help to ensure the early season advantage. Therefore, proper timing of herbicide application concerning the growth stage of the weeds is critical. Table 3 shows that the majority (66%) of the interviewed SSGs in Mona and Sonkombo indicate that chemical application (herbicides) for weed control is untimely (late), likely to result in yield decline. Eweg et al. [6] state that poor re-planting rates may contribute to reduced sugarcane yields. The results in Table 3 show that a relatively higher proportion (53%) of the interviewed SSGs have different categories of sugarcane in their fields (more than four (4) years old) indicating a challenge of late ratoon management after harvest (field clean-up). The age of sugarcane can affect yield because the higher the number of ratoon periods, the higher the demand of inputs used, especially the application of urea. Among the challenges, about 31% of the interviewed SSGs lists the cost of fertiliser as an essential factor to consider when deciding on fertiliser use and soil recommendations since they make little or no profit after harvest (Table 3). As a result of the challenges, the production of sugarcane in Mona and Sonkombo has been declining.

**Table 3.** Summary of the production challenges faced by small-scale sugarcane growers in Mona and Sonkombo (*n* = 100).

| Crop Husbandry Practice | Description of the Challenge | Percentage (%) |
|---|---|---|
| Harvesting | Late harvesting due to lack of transportation of sugarcane to the mill, immature sugarcane burning, and livestock encroachment | 100 |
| Weed control | Delay in weeding | 97 |
| Selection of seed | Lack of SSG involvement in sugarcane seed selection and the buying of seed from neighbouring growers whose seed viability is not known | 72 |
| Crop nutrition | Low and late fertilisation post or at planting | 70 |
| Chemical spray | Late/delayed chemical use or application | 66 |
| Ratoon management | Late/delayed field clean-up | 53 |
| Cost of fertiliser | Cost of fertiliser in relation to low post-harvest income | 31 |

### 3.3. Estimates of the Production Function Analysis

Before estimating the production function to determine factors affecting sugarcane production by SSGs, a correlation Pearson's coefficient is performed. In general, a correlation Pearson analysis tests the strength of the linear association between variables that is; sugarcane production (yield—t/ha) variable and the independent variables. Table 4 shows that all the five (5) variables have a statistically significant (*p* = 0.000) association with sugarcane production (yield—t/ha). These variables are InCultivated farm size (ha); Inlabour (man-days/ha); InAmount of Basal fertiliser applied (kg/ha); InAmount of Urea applied (kg/ha) and InAmount of chemicals (Gramoxone) applied (litres/ha).

**Table 4.** Correlation Pearson (point biserial) analysis results between sugarcane production (yield—t/ha) and the independent variables (*n* = 100).

| Parameter | Pearson Correlation |
|---|---|
| lnCultivated farm size (ha) | 0.994 ** |
| lnLabour (man-days/ha) | 0.644 ** |
| lnAmount of Basal fertiliser applied (kg/ha) | 0.974 ** |
| lnAmount of Urea applied (kg/ha) | 0.961 ** |
| lnAmount of chemicals (Gramoxone) applied (L/ha) | 0.730 ** |

** denotes correlation is significant at the 0.01 level (2-tailed).

Only these variables that are statistically significantly associated with sugarcane production (yield—t/ha) in the correlation Pearson analysis are included in the production function analysis specified in Section 2.5.1.

The estimates of the inferential (production function) analysis are presented in Table 5. A goodness-of-fit for the model computed for this dataset comprises the R-Square, Durbin–Watson, F-statistic, and the variance inflation factors (VIF) statistics. From the five (5) variables further considered for examination in the production function, following Pearson's correlation analysis; two (2) are inputted in the final model due to collinearity diagnostics. Only the variables with a VIF ($\leq 4$) remain in the final model. The following variables are removed because of multicollinearity; lnFarm income, lnCultivated farm size, lnAmount of basal fertiliser and lnAmount of urea applied. Using the R-Square and Adjusted R-Square coefficients to determine how well the final model fits the data, an R-Square and Adjusted R-Square of 0.629 and 0.621 are obtained, respectively (Table 5). The R-Square and Adjusted R-Square measure the "model quality" or the percentage of the variance of the results that is explained by the model. Concerning this dataset, the R-square accounts for about 63% of the variation of the dependent variable by the explanatory variables, suggesting that the model is fit to explain the variations to the dependent variable. The coefficient of the adjusted R-Square indicates that about 62% of the factors are from the hypothesised explanatory variables. Generally, the closer to one (1) the adjusted R-Square is, the better the fit of the estimated regression line. The Durbin–Watson tests value for the final model is 2.210, suggesting that there is no autocorrelation. Dlamini and Masuku [34] state that the F-statistic explains the relationship between the dependent and independent variable. Table 5 shows that an F-statistic 82.122 has a *p*-value of 0.000. Since our *p*-value ($\leq 0.05$), we reject the null hypothesis and conclude that the parameters are jointly statistically significant [35]. Therefore, this implies a statistically significant relationship between sugarcane production (yield—t/ha) and the predictor variables. Table 5 shows that the two (2) independent variables (lnLabour (man-days/ha) and lnAmount of chemicals (Gramoxone) applied) are statistically significantly affecting sugarcane production (yield—t/ha).

The variable lnLabour (man-days/ha) is statistically significant at 1% significance level (*p*-value = 0.000) and positively correlated with sugarcane production by SSGs with a regression coefficient of 0.383. The model predicts that a 1% increase in the amount of labour input (man-days/ha) will increase sugarcane production (yield—t/ha) by 0.383 per cent. The finding is in agreement with the expected outcome and also in harmony with results by, for example, Ogwang [33] and Narayan [36] who found that labour is positively correlated to sugarcane production.

The variable lnAmount of chemicals (Gramoxone) applied (litres/ha) is statistically significant at 1% significant level (*p*-value = 0.000) and positively correlated with sugarcane production by SSGs with a regression coefficient of 0.528. The model predicts that a 1% increase in the amount of chemicals (Gramoxone) applied would be associated with a 0.528 per cent increase in the sugarcane production (yield—t/ha) by SSGs. This finding is in harmony with the expected outcome and with the conclusions of Dlamini and Masuku [34] who found that the amount of chemicals applied is positively associated to agricultural productivity, provided that proper chemical spray programmes are followed.

**Table 5.** Estimates of the production function analysis (*n* = 100).

| Parameter | Coefficient (βi) | Std. Error | *t*-Statistic | Prob. (*p*-Value) | VIF |
|---|---|---|---|---|---|
| Intercept (constant) | 0.601 | 0.419 | 1.434 | 0.155 | - |
| InLabour (man-days/ha) | 0.383 *** | 0.077 | 4.992 | 0.000 | 1.371 |
| InAmount of chemicals (Gramoxone) applied (L/ha) | 0.528 *** | 0.071 | 7.484 | 0.000 | 1.371 |
| Number of observations | | | 100 | | |
| R-square | | | 0.629 | | |
| Adjusted R square | | | 0.621 | | |
| Durbin–Watson | | | 2.210 | | |
| F-statistic | | | 82.122 (*p* = 0.000) | | |

*** denotes significant at 1% level.

## 4. Discussion

Emana and Gebremedhin [24], states that the relationship between farm size and crop production (yield—t/ha) cannot be pre-determined. Instead, production may be affected by the inputs costs, and poor replant rates [23]. The finding in this paper is that SSGs in Mona and Sonkombo mainly cultivate sugarcane on small pieces of land. Wongnaa [14] and Ntabakirabose [37] found a statistically significant and negative correlation between farm size and agricultural production. Dlamini and Masuku [34] found a statistically significant and positive correlation between farm size and agricultural production. Our initial hypothesis in this paper was that the cultivated land size could have either a positive or a negative influence on sugarcane production (yield—t/ha) by SSGs. The correlation Pearson analysis shows that farm income is statistically significant and positively associated with sugarcane production (yield—t/ha) by SSGs. However, due to multicollinearity detected for this variable, it is dropped for further analysis from the production function model.

It is clear that labour, fertiliser, urea and chemicals (Gramoxone) application constitute significant production costs in the sugar industry by SSGs. The finding in this paper is that the SSGs interviewed are somehow applying urea and chemicals (Gramoxone) in line with the industry estimated standards save for basal fertiliser. Low fertiliser application and its associated costs are among the constraints pointed out by the interviewed SSGs. However, although the interviewed SSGs applied urea and chemicals (Gramoxone) in line with the industry estimated standards, the challenge is that they do not apply these in a timely fashion. The SSGs interviewed report applying urea and chemicals (Gramoxone) late, resulting in poor yield growth and weed control, respectively. The Pearson correlation analysis shows that labour, the amount of basal fertiliser, urea and chemicals (Gramoxone) applied are statistically significant and positively associated with sugarcane production (yield—t/ha) by SSGs. However, due to multicollinearity, the amount of basal fertiliser and urea applied are dropped for further analysis in the production function model. In this paper, we find labour and the amount of chemicals (Gramoxone) applied to be statistically significant and positively correlated with sugarcane production (yield—t/ha) by SSGs. This finding is consistent with Dlamini and Masuku [34], who also found that labour is statistically significant and positively correlated with sugarcane production. Similarly, Chepng'etich et al. [38] found that hired labour has a positive and statistically significant correlation with sorghum technical efficiency. In contrary, Wongnaa [14] found the labour variable to be negatively correlated with agricultural (cashew) production. The same study by Wongnaa [14] also found that chemicals (pesticide) use is positively correlated with cashew output although Dlamini and Masuku [34] did not find a statistically significant correlation between chemicals application and sugarcane production.

## 5. Conclusions

The purpose of this paper was to determine the factors affecting sugarcane production by SSGs in Mona and Sonkombo in Ndwendwe. The results show that the statistically significant factors affecting

sugarcane production (yield—t/ha) are labour and chemicals (Gramoxone) use. The descriptive findings of the paper reveal that most of the interviewed SSGs cultivate sugarcane on relatively small pieces of land. However, the finding of this paper from the production function fails to provide evidence that SSG cultivated land size is statistically significantly associated with sugarcane production. An analysis of the production challenges shows that SSGs are mainly constrained by late harvesting, a problem that could be attributed to the observed negative returns (low sugarcane production—yield). However, the declining production of SSGs is also compounded by poor agronomic practices such as inadequate and late fertiliser application, late chemical application (poor weed control), poor ratoon management, lack of involvement in seed variety selection and costs of fertiliser which could jointly result in reduced yield. However, the effect of the amount of basal fertiliser and urea applied on sugarcane production (yield—t/ha) in the production function analysis could not be established due to multicollinearity. From the production function, it can be concluded that if SSGs are to improve their production (yield) and thus maximise income, timely and adequate application of inputs, for example, chemicals (Gramoxone), is paramount. It is recommended that for SSGs to realise good yields per hectare, ideal amounts of basal fertiliser and chemicals be used according to soil testing requirements. Timely and adequate input use will undoubtedly save costs but improve production and efficacy. Additionally, out-grower technical services need to be strengthened to assist SSGs in dealing with agronomic production challenges. A considerate effort will have to be taken by SSGs to avoid late harvesting due to lack of transport and minimising fertiliser costs. One viable option will be through collective action such as agricultural cooperatives that will assist SSGs to take advantage of the buying consortiums to reduce both fertiliser and transport costs to the mills.

Further research of increased scope in terms of an increased sample size could be extended to other sugarcane growing areas to enhance the reliability and efficacy of the findings. Our study generally focused on technical efficiency; future research can also encompass allocative efficiency in sugarcane production. In the current production function, another limitation is that we are not able to capture the effect of climate change to sugarcane production.

**Author Contributions:** N.S.Z. performed the conceptualisation of the study, methodology, and investigation under the principal supervision of M.S. and B.S.T. (who co-supervised the study). N.S.Z. was responsible for the writing—original draft article preparation under the supervision of M.S., who also performed the data analysis for this paper. M.S. undertook data validation, rewriting—editing and critical review.

**Funding:** This research received no external funding.

**Acknowledgments:** Exceptional thanks goes to the SSGs who took the time to participate in this study. Their cooperation is much appreciated. Gratitude also extends to the Cane-Growers office in Mount Edgecombe for the motivational support.

**Conflicts of Interest:** The authors declare no conflict of interest.

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
