# Peer review of "Factors Affecting Sugarcane Production by Small-Scale Growers in Ndwedwe Local Unicipality, South Africa"

_agriculture, doi:10.3390/agriculture9080170_

Round 1

Reviewer 1 Report

I suggest a small changes which are highlighted directly in manuscript.

Author Response

.

Reviewer 2 Report

The use of â€˜in the study area’ is almost obsessive in all the text.

 Affiliations 1 and 2 are the same, therefore don’t need to be repeated.

Keywords small-scale growers, sugarcane, production and Ndwedwe are already cited in the title and should be replaced by other keywords

Author Response

.

Reviewer 3 Report

It is the study about sugarcane production, factors affecting its yield, study focused on small scale growers. Although study has been managed well but lacking novelty. It is study of local importance lacking any scientific soundness. 

Detail section wise comments can be seen in attached reviewed copy. 

Author Response

.

Round 2

Reviewer 3 Report

Dear Authors, 

My suggestion about socio-economics and other related factors is that parameters like Education status, Extension support, access to credit and farm income should be consider in this study as they have significant role as well like production factors while others may be skipped. So, Please write accordingly in result and discussion section.

Cultivated farm size is included in the table then why text has been deleted in the manuscript. 

Line 728; Recommendation for sugarcane growers is still missing. 

Line 732, future research directions should merge with conclusion section at the end. 

Specific comments can be seen in attached revised file.

Author Response

.
